# Perspectives of Stakeholders on Technology Use in the Care of Community-Living Older Adults with Dementia: A Systematic Literature Review

**DOI:** 10.3390/healthcare7020073

**Published:** 2019-05-28

**Authors:** Leonieke C. van Boekel, Eveline J.M. Wouters, Bea M. Grimberg, Nardo J.M. van der Meer, Katrien G. Luijkx

**Affiliations:** 1Department Tranzo, Tilburg School of Social and Behavioral Sciences, Tilburg University, 5000 LE Tilburg, the Netherlands; e.j.m.wouters@uvt.nl (E.J.M.W.); K.G.Luijkx@tilburguniversity.edu (K.G.L.); 2Health Innovations and Technology, Fontys University of Applied Sciences, School of Allied Health Professions, 5631 BN Eindhoven, the Netherlands; 3Healthcare organization Azora,7061 AP Ter Borg, the Netherlands; B.Grimberg@azora.nl; 4Department of Anesthesiology, Amphia Hospital 4818 CK Breda, the Netherlands; N.vandermeer@tias.edu; 5TIAS School for Business and Society, Tilburg University, 5037 AB Tilburg, the Netherlands

**Keywords:** dementia, older adults, technology, perspectives, informal caregivers, formal caregivers

## Abstract

Although technology has the potential to promote aging in place, the use of technology remains scarce among community-living older adults with dementia. A reason might be that many stakeholders are involved who all have a different perspective on technology use (i.e., needs, wishes, attitudes, possibilities, and difficulties). We systematically searched the literature in order to provide an overview of perspectives of different stakeholders on technology use among community-living older adults with dementia. After selection, 46 studies were included. We mainly found perspectives of informal caregivers and, to a lesser extent, of persons with dementia and formal caregivers. Perspectives of suppliers of technology were not present. Shared perspectives among persons with dementia and informal and formal caregivers were, among other things, ease of use, stability and flexibility of technology, importance of privacy, and confidentiality. We also found that among older persons, fun and pleasure, in addition to enhancing freedom and independence, facilitates technology use. Informal caregivers’ peace of mind and relief of burden also appeared to be important in using technologies. Formal caregivers value the potential of technologies to improve monitoring and communication. Insight in shared, and conflicting perspectives of stakeholders are essential to enhance the use of technology.

## 1. Introduction

Older adults prefer to live independently and to stay in their own home if possible, also referred to as ‘aging in place’ [1,2]. Aging in place is not only preferred by older adults themselves, but also encouraged by policy makers, because of the increasing number of older people within Western societies, the shortage of health care professionals, and the increase of healthcare costs [3]. However, older adults may experience difficulties in performing a variety of home maintenance tasks [4], especially when having cognitive impairments. Smart homes and technologies are often proposed as solutions for promoting aging in place [5,6]. There are various technologies that all have the potential to meet specific unmet needs of persons with dementia and their informal caregivers. Technologies may be useful in monitoring older adults with dementia in order to improve quality of life, promote physical independence, or to reduce caregiver burden [5,7,8,9]. For instance, GPS technologies may stimulate older adults with dementia to get outside more often since they enhance feelings of safety and may reduce fear or anxiety. Another example is technologies to monitor and ensure home safety such as sensors and alarms (‘remote monitoring’). These technologies are useful for risk reduction and consequently evoke feelings of safety among persons with dementia themselves, but also among their (in)formal caregivers. Other technologies can specifically help older adults with dementia to maintain functional knowledge of their personal details and of the reality around them, or provide memory training in order to retrieve information about daily activities. Finally, technologies may provide general and personalized information, support with regard to dementia symptoms, social support and company, enhancing physical activity, or health monitoring and perceived safety [10,11]. Despite the potential of technologies to facilitate and enhance aging in place for older adults with dementia, evidence for the effectiveness of technology use remains scarce and technology use often fails or is not sustainable in the long term [5,12,13,14,15]. 

Using technologies in the care of community-living older adults with dementia appears to be difficult. Dementia is a complex disorder; manifestation and progression can vary greatly, and the condition is poorly characterized and understood as well as unpredictable. All of this complicates the use of technologies [12]. Another explanation for the difficulties in using technologies for community-living older adults with dementia is the fact that many different stakeholders are involved. Older adults with dementia themselves are an important stakeholder, but many other stakeholders are involved, such as informal caregivers, formal caregivers, managers of healthcare organizations, and suppliers of technologies [16,17,18,19]. All of these stakeholders play a role in using technologies and, naturally, all have different needs, wishes, attitudes, knowledge, expectations, and experiences in this process. The Triple-I model can be useful to disentangle the differences in the perspectives of stakeholders and unravel the complexities of using technologies. According to this model, stakeholders have different identities or intrinsic values, different interests, and different ideals that play a role in using technology [20,21]. For successful use of technologies, it is essential that stakeholders who are involved interact with each other in order to achieve mutual understanding and cooperation. However, there is a lack of understanding and insight in the mutual and different perspectives of the stakeholders involved in technology use for community-living older adults with dementia [17,19]. 

The objective of our study was to provide an overview about what is known in the scientific literature about the perspectives of the different stakeholders who are involved in using technologies for community-living older adults with dementia. We define perspectives as the needs, wishes, attitudes, possibilities, and difficulties of stakeholders regarding technology use. The research questions of this study were (1) what is known about the similarities and differences in perspectives between the relevant stakeholders concerning using technologies among community-living older adults with dementia? And (2) what is the influence of the various perspectives of stakeholders on the successful use of technologies in the care for community-living older adults with dementia? 

## 2. Materials and Methods 

The databases PubMed, Web of Science, PsycINFO, Sociological Abstracts, and Sociological Services Abstracts have been systematically searched for articles published in English or Dutch since 2006. We searched a broad diversity of databases in order to cover literature from different fields such as biomedical (PubMed), psychological (PsycINFO), and social and behavioral sciences. We combined three groups of search terms, namely, (1) stakeholder perspective, (2) technologies, and (3) dementia. Table 1 shows the specific search terms we used per group combined with “OR” between the search terms or synonyms. The groups of search terms were combined by “AND” in order to find articles that focus on stakeholder perspectives as well as technology and dementia. The search strategy was identical for each database, and the final search was conducted in March 2017. 

Figure 1 displays a flowchart of the selection process. The search process yielded 1391 articles after removal of duplicates. In the first selection phase, titles were screened to see whether articles concerned stakeholder perspectives in the use of technology in dementia care. Articles of which the reviewer (BG) was uncertain proceeded to the next selection phase. In the second selection phase (*n* = 187), abstracts were judged by two independent reviewers (BG, EW, KL, and NM) on the following criteria: (1) at least one stakeholder is involved in the study, (2) the study is about technology, (3) the study concerns persons with dementia living at home, and (4) empirical research (e.g., no reviews or commentaries). Disagreements were resolved by discussion to reach consensus or when unresolved went on to the last selection phase. In this third selection phase (*n* = 100), articles were judged based on the inclusion and exclusion criteria by reading the full text, again by two independent reviewers (all authors). 

Table 2 shows the specific inclusion and exclusion criteria and definitions that were used during the selection process. We were searching for studies about the perspectives (needs, wishes, attitudes, possibilities, or difficulties in using technologies) of different stakeholders on the use of technology for community-living older adults with dementia. Since we were also interested in attitudes and opinions regarding technology, which the stakeholders did not always actually use or have experience with, sometimes the technology was only presented or described in a study as opposed to real-world use. Stakeholders who we considered to be involved in the process of technology use among persons with dementia were as follows: persons with dementia themselves, informal caregivers, formal caregivers (e.g., nursing staff, general practitioners (GPs), physicians, and home care staff), management of healthcare organizations, and suppliers of technology. We were interested in studies that focused on technologies aimed at community-living older adults with dementia to maintain independence or enhancing quality of life. Studies focusing on a specific technology (e.g., internet platform or GPS monitoring) as well as studies concerning various technologies or technology in general were all included. The technology in question did not have to be primarily or exclusively focused on persons with dementia themselves. For instance, an internet intervention to reduce burden among informal caregivers of persons with dementia was included since it was related to community-living older adults with dementia. Studies that were limited to the description of the development phase or needs assessment concerning a technology without actual use were excluded. In case a study concerning community-living older adults with dementia as well as in an institutional setting, studies were only included when the majority was living at home or, instead, we limited data extraction to results about the first and not the latter. We only included empirical studies. If a study contained a literature review as well as empirical data, we only extracted data and results from the empirical section.

The following data were extracted for each study: author, year, journal, originating country, stakeholder(s) included in the study, description of the technology or description of the measurements, description of the setting or care situation, description of the target group and/or stage of dementia, description of the perspectives of the stakeholders on technology use, (if relevant) description of the differences in perspectives of stakeholders, and (if relevant) outcomes of the intervention. The quality of the studies was assessed using the mixed methods appraisal tool (MMAT) [22]. The MMAT is a unified quality assessment tool for the appraisal of qualitative as well as quantitative and mixed methods studies. Data extraction and quality appraisal were performed by pairs of two independent reviewers (LB, BG, KL, and EW). Disparities were resolved by discussion between the reviewers, consulting a third reviewer, or by consulting the authors of the original studies. 

## 3. Results

### 3.1. General Findings

Appendix A provides an overview of the features of the primary studies. Of the 46 studies [9,23,24,25,26,27,28,29,30,31,32,33,34,35,36,37,38,39,40,41,42,43,44,45,46,47,48,49,50,51,52,53,54,55,56,57,58,59,60,61,62,63,64,65,66,67] included, the majority were conducted in Europe (*n* = 30), seven studies in Asia, and seven studies in North America, while two were multi-country studies. Informal caregivers were the most prevalent stakeholders in the primary studies (*n* = 43). Persons with dementia themselves were also often involved in studies (*n* = 24). However, it differed per study whether data collection actually took place among persons with dementia or whether a proxy served as input for the perspective of persons with dementia. Formal caregivers were a stakeholder in 16 primary studies, varying from GPs to nurses to occupational therapists. In one study, managers were included as stakeholders, albeit as part of a larger group of formal caregivers (9 managers within a group of 96 formal caregivers) [42]. Considering the small number of managers in the primary studies, the perspective of managers will not be reported separately, but included among the perspectives of the formal caregivers as a whole. The perspectives of suppliers of technology were not present in the included studies, although we included them in the initial search string.

In 28 studies, a technology was actually used by participants of the study, and 18 studies focused on stakeholders’ attitudes, opinions, or expectations regarding technology in general or a technology that they did not (yet) actually use. The technologies investigated in 28 studies were heterogeneous; technologies providing support to informal caregivers (e.g., information, peer-to-peer contact, and personalized advice) *n* = 6, assistive technologies (e.g., alarms, sensors) *n* = 6, intervention program via internet or telephone *n* = 4, GPS monitoring system *n* = 3, technology to facilitate or improve communication between caregivers and/or persons with dementia *n* = 3, technology providing support for persons with dementia (e.g., reminders, picture dialing, information about hometown) *n* = 3, monitoring system in the home of the person with dementia (e.g., several sensors and detectors within the home) *n* = 2, and simple remote control *n*=1. The 18 studies investigating attitudes, opinions, or expectations with regard to a technology predominantly involved informal caregivers. In these studies, persons with dementia were not included. 

The perspectives we found in the studies are reported separately per group of stakeholders and as shared perspectives. If a perspective was particularly relevant for one category of technologies, it was highlighted. In general, most perspectives were applicable for various technologies. Appendix A provides an overview of the main findings per study sorted by perspectives of the stakeholder.

### 3.2. Shared Perspectives on Technology use Among all Stakeholders

Shared perspectives that were mentioned by persons with dementia as well as informal and formal caregivers are the importance of ease of use and having a sense of capacity to use the technology [9,30,31,35,37,47,48,49,53]. Being worried about user-friendliness appeared to be a barrier for technology use. Furthermore, the stability and flexibility of the technology appeared to be important for all stakeholders. Technologies that are not functioning as intended cause frustration, and it takes time and energy to solve possible problems. In addition, unstable technologies may cause loss in confidence and reliability perception that can be a barrier to continue use [9,43,48,51]. The importance of privacy and confidentiality of the technology were mentioned by all stakeholders [35,40,49]. Especially in GPS technologies, this perspective appeared to be important [33,40,53,61]. A facilitator for technology use among all stakeholders was the fact that a technology could be easily incorporated in daily life and habits [36,43,51,62]. Finally, the timing of the introduction of the technology with regard to disease progression appeared to be crucial [9,28,29,34,36,37,40,52,53]. When a technology is introduced too early, the person with dementia may not take it seriously. When a technology is introduced rather late, it is difficult for persons with dementia to understand and to get used to, which in turn sometimes makes it more difficult to incorporate the use of the technology in a patient’s daily life. 

### 3.3. Perspectives on Technology Use among Persons with Dementia

Facilitators for the use of technology among persons living with dementia are the potential of technology to maintain or enhance their freedom and independence and subsequently allow them to live longer at home or postpone institutionalization [33,46,49,53,58]. The use of technology among persons with dementia may also evoke positive emotions such as a feeling of mastery and the feeling of being digitally included [24,43]. In addition, having fun and pleasure are facilitators for technology use [31,32,52]. Several facilitators for the use of technology correspond to the aim of the technology. For instance, GPS evokes a feeling of safety and security, which was found to be a facilitator for persons with dementia to use GPS [33,46,58]. In addition, technologies for communication purposes stimulate social interactivity or enhance support for persons with dementia, which was positively evaluated [54]. In sum, the aforementioned facilitators for technology use among persons with dementia are, in most cases, in line with the purposes of the technology. 

In the development and design of technologies, there are a few points of interest to keep in mind since they play a role in the use of technology for persons with dementia. First, the visibility and aesthetics of the technology. The technology should not be stigmatizing or embarrassing, since this may be a barrier for persons with dementia to use it [33,34,46,61]. Persons with dementia may also have certain concerns that are important to keep in mind in using technologies, such as the costs of technologies, the possibility of breaking the technology, or a heightened vulnerability to criminals [47,52,53,61]. In addition, noises and lights might be confusing for persons with dementia and may therefore be a barrier in the use of technology [9,34,60]. 

### 3.4. Perspectives on Technology Use among Informal Caregivers

Informal caregivers are in favor of using technologies when they see potential and positive effects for the person with dementia. For instance, technologies that enhance freedom and independence for persons with dementia are evaluated positively [33,36,40,46,58,66]. Furthermore, informal caregivers value the effect that technology can have on quality of life or quality of healthcare delivery for the person with dementia [55,61,66]. Informal caregivers attach great value to risk reduction, protection, and safety for persons with dementia as a consequences of technology use [35,40,46,66]. On the other side of the coin, informal caregivers sometimes attach less value to consequences of technology use on liberty and autonomy of the person with dementia [49,59,61,66]. 

Informal caregivers also mention specific reasons for using technologies that have positive effects for themselves. For instance, technologies have the potential to reduce informal caregivers’ level of stress, increase peace of mind, and reduce their worries about the person with dementia [26,28,35,40,42,61,65]. Technology can also provide them more freedom, save them time, and provide them with a relief from a burden that was evaluated positively [24,34,35,47,49,58,63,66]. Some technologies can also provide support for informal caregivers, such as provision of information, and increase their confidence level or peer-to-peer contact [44,63,65]. In addition, technologies that improve the relationship between the informal caregiver and person with dementia are positively evaluated. For instance, doing something together, having less conflicts, or improving communication [32,38]. 

Informal caregivers mentioned potential barriers for the person with dementia in using technology. These are in line with barriers mentioned by persons with dementia themselves. For instance, sounds and lights may be confusing or cause anxiety; there are also the aforementioned concerns about privacy and vulnerability for criminals [59,61,63]. Informal caregivers perceive costs of technologies and time commitment to learn to use the technology or to solve problems with the technology as barriers in technology use [30,47,59]. Consequently, technical support or assistance from formal caregivers in technology use can be a facilitator for technology use [44]. In addition, informal caregivers mention the importance of the flexibility, stability, and simplicity of the technology [62]. Lastly, it is important that it is easy to learn to use the technology and that it is useful for multiple users [30]. 

### 3.5. Perspectives on Technology Use among Formal Caregivers

Formal caregivers value the potential of technologies to improve documentation and monitoring of their patients [63]. This provides the opportunity to react timely on status changes. In addition, improvement of interaction between them and informal caregivers and persons with dementia by the use of technologies was positively evaluated [63]. Technologies can also save (travel) time and as a consequence save costs for formal caregivers [25,35,52]. However, costs can also be a barrier for formal caregivers to use technologies [35,51]. Among formal caregivers, there are some concerns that technologies might decrease face-to-face contact with persons with dementia or dehumanize care [35]. Additionally, less involvement of family and reduced personal contact or more distance in the relationship (between informal and formal caregivers) were mentioned as a potential barrier [40,64]. However, the opposite can also hold true, where technology has the potential to actually improve the relationship between formal and informal caregivers, or between informal caregivers and their peers [64]. Especially in rural areas, technologies might be useful to enhance support and contact between informal and formal caregivers [63]. Finally, privacy issues are also a concern among formal caregivers [35,40,42].

## 4. Discussion

In this literature review, we predominantly found perspectives of informal caregivers on technology use and, to a lesser extent, perspectives of persons with dementia and formal caregivers. The perspectives of suppliers of technology was not present in the included studies. Shared perspectives among all stakeholders were, among other things, the ease of use, stability, and flexibility of technology, and the importance of privacy and confidentiality. Persons with dementia value the potential of technology to have fun and pleasure with it as well as its potential to enhance their freedom and independence. Among informal caregivers, having peace of mind and relief of burden were important facilitators for technology use. Formal caregivers appreciate the fact that technologies may improve monitoring of their patients and interactions with other stakeholders. Although we specifically searched for perspectives of stakeholders on technology use among community-living older adults with dementia, it appears that the perspectives we found are to a large extent in line with previous findings among older adults without dementia [19,68].

In some cases, we found conflicting facilitators and barriers for the use of technology among community-living older adults with dementia. For instance, informal caregivers as well as persons with dementia value the fact that technology, especially GPS solutions, enhance their freedom and independence. Nevertheless, in this case it is always known where persons with dementia are located, which reduces their privacy. The same applies to monitoring systems such as alarms, sensors, or cameras within the home of persons with dementia. Formal caregivers value the fact that technologies can enhance contact with their patients and their informal caregivers. However, dehumanizing care and less face-to-face contact is a consequence of technology use that was mentioned as a potential barrier. Insight into the different perspectives of the stakeholders is important to prevent conflicting or contradictory perspectives from becoming a barrier to using technologies. For instance, to prevent privacy issues from becoming a barrier to technology use, it is important to be clear about who has access to data, how data are stored, and how they are used. In addition, it is important to agree who is responsible to act upon signals or problems [69].

The timing of the introduction of technologies appears to be crucial, since it came up in several studies among various stakeholders and various technologies. As mentioned before, dementia is a complex disease that comes with different (sometimes distinct, sometimes gradual) phases. The same applies for the available technologies; some are specifically designed for the first phases of dementia, such as GPS technologies or memory assistive technologies. Other technologies better suit more advanced stages of dementia, such as sensory stimulating technology or monitoring systems within the home. When introducing a technology, it is therefore vital that it matches the phase of dementia. Additionally, among community-dwelling older adults who do not suffer from dementia, timing of the introduction of technology is essential [68]. As suggested by Nijhof and colleagues [70,71], besides specific technologies for every phase of dementia, the involvement of stakeholders is different per phase of dementia. In the first phase of dementia, persons with dementia themselves play a major role, since they are still relatively independent and able to learn. In more advanced stages of dementia, (in)formal caregivers become more important in the use of technologies. Since stability and flexibility of the technology, as well as timing of the introduction of the technology, appear to be of great importance for successful use, it might be worthwhile for suppliers to make it possible to rent or lease technologies for community-living older adults. This would make technology use more flexible; it may be more cost-effective and therefore more feasible to introduce technologies at the right moment within the disease progress.

The results of our study should be considered in the light of some limitations. First, the majority of the included studies in this literature review are from Western countries. It is known that social support, coping, and the dementia caregiving experience are largely affected by race, ethnicity, and culture [72]. Therefore, the reported perspectives of the stakeholders may not be, or only partially, applicable to non-Western societies. Secondly, we only included ‘perspectives’ in our search string. It might have been better to include search terms that capture perspectives such as wishes, needs, attitudes, possibilities, or difficulties. However, we explicitly included the different stakeholders in our search terms, and therefore we expect that we have found most studies that capture perspectives of stakeholders. Another limitation is the fact that in some studies, it was unclear whether perspectives were the actual opinions of persons with dementia or whether (in)formal caregivers thought it might be of importance for persons with dementia. Nowadays, there are research methods and knowledge available about how to ask, observe, or involve persons with dementia themselves instead of asking a proxy [73,74,75]. Since the perspectives of the stakeholders were sometimes different, especially the perspective of informal caregivers, is it important to include persons with dementia themselves to unravel their perspectives on technology use.

This literature review provides an extensive overview of perspectives of persons with dementia and their informal and formal caregivers on a broad variety of technologies. It appeared that the perspectives on technology use are to a great extent comparable with findings among community-living older adults without dementia or cognitive problems. There is a paucity of knowledge in academic literature about the perspective of suppliers of technology. This study provides insight into whether perspectives of people with dementia and their informal and formal caregivers correspond or are differ with each other. This knowledge is important, since it may influence, impede, or enhance technology use among community-living older adults with dementia. 

## 5. Conclusions

This literature review provides an extensive overview of perspectives of persons with dementia and their informal and formal caregivers on a broad variety of technologies. It appears that the perspectives on technology use are to a great extent comparable with findings among community-living older adults without dementia or cognitive problems. There is a paucity of knowledge in academic literature about the perspective of suppliers of technology. This literature study has some practical implications. The themes described are, in general, applicable to a variety of technologies for community-living older adults. In addition, the findings provide insight into perspectives of people with dementia and their informal and formal caregivers regarding technology use. As aforementioned, this insight regarding perspectives of involved stakeholders is crucial since it may influence, impede, or enhance technology use among community-living older adults with dementia. Successful use of technology in complex situations such as dementia care, with multiple stakeholders, requires acknowledgement of the perspectives of all these stakeholders.

## Figures and Tables

**Figure 1 healthcare-07-00073-f001:**
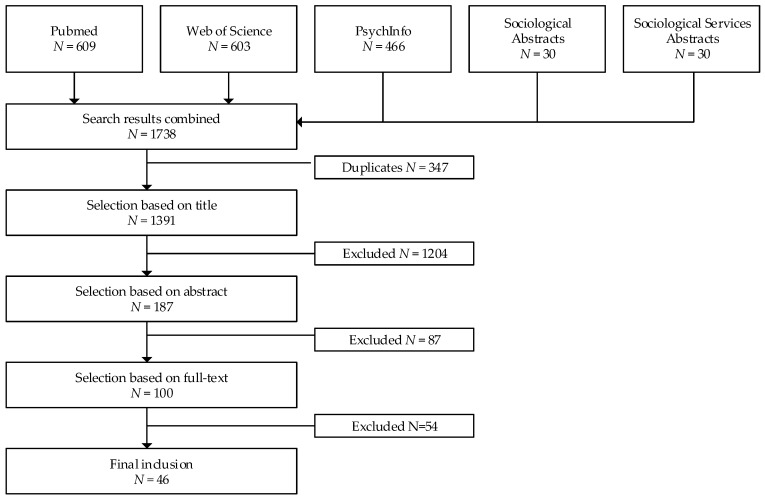
Flowchart of the selection process.

**Table 1 healthcare-07-00073-t001:** Groups of search terms.

**1. Stakeholder perspective**	
Stakeholder(s)	Organization(s)
General practitioner(s)	Client(s)
Caregiver(s)	Patient(s)
Care professional(s)	Elderly
Supplier(s)	Elderly people
Provider(s)	Older people
Management	Different perspectives
Manager(s)	
**2. Technology**	
Ehealth/e-health	Telemedicine/tele-medicine
mhealth/m-health	Assistive technology
Robotics	Assisted living
Robotic technology	Technology acceptance
Sensor-based networks	Technology adoptation
Domotics	Tele-monitoring/telemonitoring
Smart home(s)	Electronic tracking
Home automation	Sensor technology
Care technology	Gero(n)technology
Telecare/tele-care	
**3. Dementia**	
Dementia	Alzheimer’s disease
Alzheimer	Alzheimers disease

**Table 2 healthcare-07-00073-t002:** Inclusion and exclusion criteria.

**Inclusion Criteria**
Studies that investigate a perspective (needs, wishes, attitudes, possibilities, or difficulties) towards technology use; Subject of the study is at least one stakeholder involved in the use of technology (persons with dementia, informal caregivers, formal caregivers, (management) of healthcare organization, or suppliers of technology); Studies concerning technology (not necessarily primarily) aimed at persons with dementia living at home to maintain independence or quality of life; Empirical studies (published in English or Dutch and after 2006)
**Exclusion Criteria**
Subject of the study is limited to (health care) students only; The majority of the included participants or groups of interest in the study are persons with dementia living in an institutional setting; Study is limited to the description of the development/pilot phase of a technology without actual use of the technology among persons with dementia; Study is limited to a needs assessment among stakeholders regarding the development of technology.

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
