# Peer review of "Perspectives of Stakeholders on Technology Use in the Care of Community-Living Older Adults with Dementia: A Systematic Literature Review"

_healthcare, 2019, doi:10.3390/healthcare7020073_

Round 1
Reviewer 1 Report
The Assistive technology including smart home, GPS etc has gained interest as a novel intervention in a range of clinical settings including people with dementia. They enhance quality of life and give more independence to people with dementia while they reduce caregivers burden. On the other hand since the global prevalence of dementia rises, care costs impose a large burden on healthcare systems and thus technology solutions in dementia care have the potential to ease this burden as well. The review by van Boekel et al on the technology use in the care of community-living older adults with dementia is very interesting; the authors analyze several perspectives in regards to (in)formal caregivers and the people with dementia. The authors have systematically searched the literature and have tried to summarize all the benefitsl by the use of technology but they also underline the complexity of the whole issue in regards to efficiency for people with dementia and stakeholders. Ethical issues have also arised. The paper is well organized and written and the introduction is well-justified. Although, as the author state, there are several limitations the review is interesting και innovative and highlights the potential novel sart technologies offer for improving the quality of life of dementia patients and caregivers but further effort is needed to provide formal guidance in order to address both benefits and potential harms.
The paper could be publish as it stands
Author Response
The Assistive technology including smart home, GPS etc has gained interest as a novel intervention in a range of clinical settings including people with dementia. They enhance quality of life and give more independence to people with dementia while they reduce caregivers burden. On the other hand since the global prevalence of dementia rises, care costs impose a large burden on healthcare systems and thus technology solutions in dementia care have the potential to ease this burden as well. The review by van Boekel et al on the technology use in the care of community-living older adults with dementia is very interesting; the authors analyze several perspectives in regards to (in)formal caregivers and the people with dementia. The authors have systematically searched the literature and have tried to summarize all the benefitsl by the use of technology but they also underline the complexity of the whole issue in regards to efficiency for people with dementia and stakeholders. Ethical issues have also arised. The paper is well organized and written and the introduction is well-justified. Although, as the author state, there are several limitations the review is interesting και innovative and highlights the potential novel sart technologies offer for improving the quality of life of dementia patients and caregivers but further effort is needed to provide formal guidance in order to address both benefits and potential harms.
The paper could be publish as it stands
Response 1: We would like to thank the reviewer for the compliments regarding our manuscript.
Reviewer 2 Report
INTRODUCTION: Although this section is well written, some key papers are missing E.g.
Exploring factors that impact the decision to use assistive telecare: perspectives of family care-givers of older people in the United Kingdom; ERICA J. COOK et al 2017; and The acceptability of assistive technology to older people; McCreadie and Tinker 2005
METHODS: This section is very clear, apart from the inclusion/exclusion criteria as set out in Table 2 where some information is missing. E.g. what time period is covered by the review; was it limited to papers in English?
RESULTS: If available, it would be interesting to report on which technologies were mentioned in relation to each theme. e.g. which technologies were said to reduce stresss for caregivers?
Finally, the manuscript would benefit from the addition of a conclusion, including implications for practice.
Author Response
Response to Reviewer 2 Comments
INTRODUCTION: Although this section is well written, some key papers are missing E.g.
Exploring factors that impact the decision to use assistive telecare: perspectives of family care-givers of older people in the United Kingdom; ERICA J. COOK et al 2017; and The acceptability of assistive technology to older people; McCreadie and Tinker 2005
Response 1: We would like to thank the reviewer. We agree that references are missing about which stakeholders are involved in technology use among older adults and papers that are investigating differences in acceptability among different stakeholders. Therefore we included the references mentioned by the reviewer (Cook et al. 2018 and McCreadie et al. 2005) as well as some other papers (Sponselee et al. 2008 and Peek et al. 2016) which have studied acceptability of various stakeholders (line 66).
METHODS: This section is very clear, apart from the inclusion/exclusion criteria as set out in Table 2 where some information is missing. E.g. what time period is covered by the review; was it limited to papers in English?
Response 2: We now added that we only included articles in English and Dutch since this was missing (line 85). We only included empirical studies published between 2006 and March 2017 (see line 85). To be more clear we now also included this information in table 2.
RESULTS: If available, it would be interesting to report on which technologies were mentioned in relation to each theme. e.g. which technologies were said to reduce stresss for caregivers?
Response 3: We agree with the reviewer that this would be interesting. Therefore in appendix B we mentioned the themes as well as which technology was at the center of the study.
The objective of our literature study was to provide an overview about what is known about the perspectives of the different stakeholders who are involved in using technologies for community-living older adults with dementia. So we were interested in a broad variety of technologies. Additionally, we actually also searched for patterns in technologies in relation to themes. However, we concluded that this was not informative and that we could not extract this information based on the studies we included in our review. First, many studies focused not on one technology but on a broad variety of technologies. Secondly, as stated in line 198-200, some themes that we found that enhance technology use are also in line with the purpose of certain technologies. For instance, ‘enhancing social interactivity’ was a theme in a study about a technology for communication purposes, which obviously is also the objective of the technology.
Nevertheless, in the results section we included information when themes were more often present among certain technologies. Themes as privacy and confidentiality seem to be more important in GPS technologies (line 80-82) and themes as such safety and security often came up in studies about GPS (see line 196-197).
Finally, the manuscript would benefit from the addition of a conclusion, including implications for practice.
Response 4: We changed the last paragraph in the discussion in order to be more explicit about implications for practice (line 309-318). However, we are aware that these implications remain quite general. The aim of our literature review was to provide an overview about what is known about perspectives of stakeholders on technology use related to and within this target group. More specific research is needed in order to investigate whether themes mentioned in our findings are applicable in specific contexts.

Reviewer 3 Report
In this work, in a logical and systematic manner was conducted on the analysis of study from the last 12 years regarding the possibility of using technologies and tools to facilitate the functioning of people with dementia staying at home. The analysis used in this article is not a typical meta-analysis of the results of various studies, however, the research selection scheme responds to the questions asked by the authors (regarding the attitudes of certain groups of stakeholders to new technologies and tools supporting person with dementia), and the conclusions obtained on this basis have practical and application character.
In such an arrangement of research, it might be a better solution to combine chapter Results with chapter of Discussion of the results, because the authors relate to a small extent to other research than those selected and analyzed in this work.
The comparing the results of the various analyzed papers would be more complete if the authors mentioned the distribution of responses to the assessed questions among specific stakeholder groups (how many people gave specific answers).
Similarly, it would be good to mention the type of technology and tools which were better accepted by specific groups of stakeholders (formal, informal caregivers, people with dementia). They were included in the description of the research used in the analysis, and were contained in the supplement, but their comparison in the text would facilitate the assessment of similarities and differences in the approach of stakeholders to use new technologies and tools to improve the quality of life of persons with dementia and their caregivers.
Author Response
Response to Reviewer 3 Comments
In this work, in a logical and systematic manner was conducted on the analysis of study from the last 12 years regarding the possibility of using technologies and tools to facilitate the functioning of people with dementia staying at home. The analysis used in this article is not a typical meta-analysis of the results of various studies, however, the research selection scheme responds to the questions asked by the authors (regarding the attitudes of certain groups of stakeholders to new technologies and tools supporting person with dementia), and the conclusions obtained on this basis have practical and application character.
In such an arrangement of research, it might be a better solution to combine chapter Results with chapter of Discussion of the results, because the authors relate to a small extent to other research than those selected and analyzed in this work.
Response 1: We would like to thank the reviewer. We agree that the discussion section is not very extensive and in some parts in line with the results. We now extended the implications for practice, see also response 3. In the result section we only focus on findings from the 46 studies included in the review. In the discussion section we also included other relevant studies that were not found in our search. This is in line with the PRISMA guideline for systematic literature reviews. In order to avoid confusion we are of the opinion that it is more logical to have a separate results and discussion section.
The comparing the results of the various analyzed papers would be more complete if the authors mentioned the distribution of responses to the assessed questions among specific stakeholder groups (how many people gave specific answers).
Response 2: Our objective was to provide insight in different perspectives of stakeholders involved in using technologies among community-living older adults with dementia. Therefore, in appendix B the reader can read the themes per study and also which specific technology and stakeholder(s) it concerned. To study opinions of one specific group of stakeholders concerning one specific technology a different study design is needed and this was not the objective of our literature study.
In response to the question of the reviewer how many people gave specific answers; since a large amount of the included studies were qualitative or mixed-method (32 studies) it was not possible to provide exact numbers or percentages of respondents agreeing to specific questions. The majority of studies involved themes and therefore comparing included studies was not possible. As also recommended in the adapted PRISMA for reporting systematic reviews of qualitative and quantitative evidence, in our results section we summarize main themes.
Similarly, it would be good to mention the type of technology and tools which were better accepted by specific groups of stakeholders (formal, informal caregivers, people with dementia). They were included in the description of the research used in the analysis, and were contained in the supplement, but their comparison in the text would facilitate the assessment of similarities and differences in the approach of stakeholders to use new technologies and tools to improve the quality of life of persons with dementia and their caregivers.
Response 3: In line with our answer to reviewer 1 response 3; in appendix B we mentioned the themes as well as which stakeholder(s) (and which technology) was at the center of the study. The objective of our literature study was to provide an overview about what is known about the perspectives of the different stakeholders who are involved in using technologies for community-living older adults with dementia. Additionally we actually searched for patterns in themes mentioned more often among certain stakeholders. However, we concluded that we could not extract this information based on the studies we included in our review. First, the distribution of stakeholders over the included studies was unequal (formal caregivers were underrepresented) and also many studies did not focus on one specific technology. Secondly, some technologies were especially aimed at one group of stakeholders (e.g. Chiu et al. 2011, support system for informal caregivers; and Cristancho-Lacroix et al. 2016 intervention to reduce stress among informal caregivers, Kerssens et al. 2015 support and entertainment system for people with dementia). It does not add relevant information to compare whether these technologies are accepted better by the stakeholders on which the technology is aimed.
However, there are other studies that specifically were designed to investigate whether certain technologies are better accepted by specific stakeholders. Also some of the primary studies of this literature review actually address these topics (e.g. Burstein et al. 2015, Mao et al. 2015).